# Dual-Attention BiLSTM for Interpretable Forecasting of Treatment Toxicities

Eric Ababio Anyimadu[1]   Xinhua Zhang[2]   Clifton David Fuller[3]   G. Elisabeta Marai[2]   Guadalupe Canahuate[1]

*Abstract*—Longitudinal patient-reported outcomes (PROs) provide crucial insights into symptom progression and treatment response in oncology, enabling more personalized and anticipatory care. Deep learning models such as Bidirectional Long Short-Term Memory (Bi-LSTM) networks have recently been applied to forecast symptom trajectories from PRO data. While these models offer improved predictive performance over traditional statistical methods, they often fall short of capturing the evolving clinical relevance of individual symptoms or the varying influence of specific time points in the patient journey and lack clinical interpretability.

In this work, we propose an attention-enhanced Bi-LSTM model that incorporates dual attention mechanisms at the item and temporal levels to selectively emphasize the most informative symptom-time interactions. This architecture addresses the limitation of uniform input weighting, enhancing both forecasting accuracy and interpretability. Evaluated on a longitudinal PRO dataset collected at a major cancer center, our model outperforms conventional Bi-LSTM approaches in predicting 12-month symptom severity and offers clinically meaningful insights into symptom evolution. These findings highlight the potential of attention-based temporal modeling to support personalized, timely decision-making in oncology care.

*Index Terms*—Longitudinal Forecasting, Patient-Reported Outcomes, Deep Learning, Dual Attention, Bidirectional LSTM, Interpretability

## I. INTRODUCTION

LONGITUDINAL study designs in oncology are essential for capturing the dynamic trajectories of disease progression, treatment response, and symptom burden over time [1]. These designs leverage time-series data such as clinical assessments, medication records, and symptom severity reports to provide a comprehensive view of a patient's journey through cancer treatment and recovery which provides an essential perspective for evaluating treatment efficacy and long-term disease control [2], [3].

Among the various sources of longitudinal data, patient-reported outcomes (PROs) have emerged as a critical tool for capturing patients' real-time experiences [4]. PROs are self-reported assessments that reflect multivariate symptom severity and quality of life before, during, and after treatment [5]. Unlike clinician evaluations, they provide a direct window into the patient's subjective experience of illness and therapy, offering insights that may otherwise be overlooked. PROs have become increasingly integrated into oncology practice, guiding treatment decisions during therapy and supporting the monitoring of recovery and late effects post-treatment [4].

However, most longitudinal studies in oncology including those involving PROs are predominantly retrospective analyses of historical cohorts with known outcomes [3], [4], [6], [7]. Although such studies are valuable for uncovering population-level patterns, they primarily support reactive care, where clinical decisions are made only after symptoms worsen or adverse events are observed. This highlights a critical gap in the ability to provide proactive patient-centered care, specifically the capacity to leverage early PRO data, collected at diagnosis or before treatment, to anticipate future symptom severity and complications. Doing so not only enables more timely and personalized treatment strategies, but also addresses the growing challenge of managing long-term effects of cancer therapy as survival rates improve [4]. Accurate forecasting of long-term symptom severity from early PRO data allows clinicians to intervene earlier, adjust treatment plans more effectively, and reduce the risk of avoidable complications, supporting a shift toward more proactive and individualized care throughout the treatment course [3].

Recently, a deep learning model approach using Bidirectional LSTM (Bi-LSTM) model has shown great promise in the prediction of PRO data [3]. However, despite their strengths, Bi-LSTM models treat all symptoms and time points uniformly, applying shared weights across the sequence that limit adaptability to contextual changes [8]. Furthermore, symptom input weights remain fixed after training, failing to reflect the evolving relevance of individual symptoms over time [9]. This rigidity not only constrains predictive accuracy but also hampers interpretability, potentially obscuring clinically meaningful patterns crucial for informed decision-making.

Clinical studies in head and neck cancer (HNC) highlight the limitations of uniform modeling. Severe early symptoms such as nausea, vomiting, and taste disturbances are often predictive of late effects such as dry mouth and swallowing difficulties. Also, symptom burden typically intensifies during treatment, peaks near its end, and gradually improves, though rarely returning to baseline. However, symptom patterns shortly after treatment often mirror those seen a year later,

This work was supported by NIH NCI-R01-CA258827, NIH UG3 TR004501, NSF CNS-2320261, and the UIC Institute for Health Data Science Research.

[1]E. Anyimadu and [1]G. Canahuate are with the University of Iowa, Iowa City, IA 52242 USA (e-mail: eric-anyimadu, guadalupe-canahuate@uiowa.edu).

[2]X. Zhang, and [2]G. Marai are with the University of Illinois Chicago, Chicago, IL 60607 USA (e-mail: zhangx, gmarai@uic.edu).

[3]C. Fuller is with the University of Texas MD Anderson Cancer Center, Houston, TX 77030 USA (e-mail: dfuller@mdanderson.org).

indicating that early and late post-treatment time points may hold similar prognostic value [10], [11], [3]. These insights underscore the need for models that can selectively attend to the most informative symptom-time combinations, enhancing both predictive accuracy and clinical interpretability.

To address these challenges, we propose enhancing the Bi-LSTM architecture with a dual attention mechanism; one to capture symptom importance and another to assess time point relevance.

We evaluate our proposed dual-attention Bi-LSTM (DAt-BiLSTM) through an ablation study, comparing it to a standard Bi-LSTM without attention to assess the impact of incorporating item and temporal attention.

Experiments are conducted using PRO dataset collected at M.D. Anderson Cancer Center using the MDASI-HN cancer questionnaire, which captures general and head-and-neck-specific symptoms.

Results show that our attention-enhanced model not only improves predictive accuracy but also yields clinically meaningful interpretability by highlighting key symptoms and time points driving each prediction.

In summary, this study introduces a Bi-LSTM model with integrated item and temporal attention to dynamically prioritize informative symptom-time pairs for long-term PRO prediction. The attention mechanisms enhance both predictive accuracy and interpretability, offering transparent insights into symptom and time point relevance and outperforming reference models on a major institutional PRO dataset.

## II. RELATED WORK

Early efforts to forecast long-term PROs explored statistical models such as Autoregressive Integrated Moving Average (ARIMA), which model each symptom independently and do not capture temporal or inter-symptom dependencies [1]. In contrast, recent advances have introduced deep learning models such as Long Short-Term Memory (LSTM) networks, which better handle multivariate time series and temporal dynamics [12], [3]. LSTMs have shown superior performance over ARIMA, with reported AUC scores of 0.75–0.89 versus 0.72–0.84 for predicting long-term symptom severity [12].

Building on this, Bidirectional LSTM (Bi-LSTM) networks have been employed to further enhance performance by processing longitudinal data in both forward and backward directions, thereby capturing more comprehensive temporal dependencies. When applied to PRO data, Bi-LSTM models have demonstrated approximately 15% lower root mean squared error compared to unidirectional LSTMs when forecasting symptom trajectories [12], [3].

To build on the strengths of Bi-LSTM models, attention mechanisms have been introduced to enhance their ability to capture nuanced patterns at both the feature and temporal levels in longitudinal data. These attention-enhanced Bi-LSTMs have been widely applied across diverse domains to better model complex dynamics, for example, improving accuracy and generalization in electricity load forecasting [8]

and identifying key temporal patterns in sea surface temperature modeling [13]. By assigning differential weights to input features and time steps, these models selectively focus on the most informative elements.

In recent years, Transformer-based models have also gained notice for time-series forecasting due to their ability to capture long-range dependencies. However, their application in clinical longitudinal data is limited by the need for large datasets and extensive tuning, often unavailable in this domain where data are sparse and limited in scale. Additionally, Transformers tend to lack inherent interpretability without substantial engineering effort [14]. In contrast, attention-enhanced Bi-LSTM models offer greater robustness and interpretability, qualities critical to clinical applications.

Despite their demonstrated success and strengths, applications of attention-enhanced Bi-LSTMs to PRO forecasting in oncology remain limited. PRO data differ fundamentally from structured sensor inputs; they are multivariate, temporally irregular, and shaped by patient-specific variability, posing unique modeling challenges. These characteristics highlight the need for attention mechanisms that adaptively emphasize prognostically relevant features without obscuring key relationships.

In response, our DAt-BiLSTM introduces a dual attention framework tailored to PRO data. First, we apply item-level attention directly to symptoms in the input data, allowing the model to emphasize informative symptoms before temporal encoding using the Bi-LSTM [9]. Then, we apply temporal attention over Bi-LSTM outputs to focus on key time points in the symptom trajectory.

## III. METHODOLOGY

In this section, we detail our proposed framework, which combines Bi-LSTM with item and temporal attention to forecast long-term PROs from early data, aiming to improve both accuracy and interpretability.

### A. Problem Formulation

Given $N$ patients, PRO scores recorded over $T$ time points for $S$ symptom items, the longitudinal PRO data is represented as a tensor:
$$\mathbf{X} = \left[\mathbf{x}_{t_0}, \mathbf{x}_{t_1}, \ldots, \mathbf{x}_{t_{T-1}}\right] \in \mathbb{R}^{N \times T \times S}, \quad (1)$$
where each $\mathbf{x}_t \in \mathbb{R}^{N \times S}$ is the matrix of symptom scores for all $N$ patients at time $t$.

The forecasting objective is to predict PRO scores at future time points using observed data up to the current time. Starting from baseline $\mathbf{x}_{t_0}$, we recursively predict intermediate symptom vectors:
$$\hat{\mathbf{y}}_{t+1} = f\left(\mathbf{x}_{t_0}, \hat{\mathbf{y}}_{t_1}, \ldots, \hat{\mathbf{y}}_t\right) \in \mathbb{R}^{N \times S}, \quad (2)$$
where $\hat{\mathbf{y}}_{t+1}$ is the predicted symptom matrix at time $t+1$ for all $N$ patients.

### B. Proposed Model Architecture

The proposed model integrates item attention, a BiLSTM encoder, and temporal attention to forecast symptom trajectories over time as illustrated in Figure 1. This architecture

enhances both predictive performance and interpretability by enabling the model to focus selectively on the most informative symptom-time combinations.

To maintain the interpretive integrity of PRO items or symptoms, linear transformations are used to minimize distortions in subjective responses.

*1) Item Attention:* For a given input sequence $\mathbf{X} \in \mathbb{R}^{N \times T \times S}$, we apply item attention to weigh the importance of different symptoms. The item attention computes attention scores using a linear transformation:

$$\mathbf{E} = \mathbf{X}\mathbf{W}_e, \tag{3}$$

where $\mathbf{W}_e \in \mathbb{R}^{S \times S}$ is a learnable weight matrix.

The normalized attention weights are obtained by applying softmax along the item dimension:

$$\mathbf{A} = \text{softmax}(\mathbf{E}) = \frac{\exp(\mathbf{E})}{\sum_{j=1}^{S} \exp(\mathbf{E}_{:,:,j})}, \tag{4}$$

where $\mathbf{A} \in \mathbb{R}^{N \times T \times S}$ contains the normalized attention weights. The item-attended input $\tilde{\mathbf{X}} \in \mathbb{R}^{N \times T \times S}$ is obtained by element-wise multiplication:

$$\tilde{\mathbf{X}} = \mathbf{A} \odot \mathbf{X}. \tag{5}$$

*2) Bidirectional LSTM Encoding:* The item-attended sequence $\tilde{\mathbf{X}}$ is processed through a bidirectional LSTM to capture temporal dependencies:

$$\mathbf{H} = \text{Bi-LSTM}(\tilde{\mathbf{X}}) \in \mathbb{R}^{N \times T \times 2H}, \tag{6}$$

where $2H$ denotes the concatenated forward and backward hidden dimensions.

*3) Temporal Attention:* For each symptom $s \in \{1, 2, \ldots, S\}$, we apply a separate temporal attention mechanism. The temporal attention computes unnormalized attention scores using a linear layer:

$$\mathbf{u}^{(s)} = \mathbf{H}\mathbf{W}_s \in \mathbb{R}^{N \times T \times 1}, \tag{7}$$

where $\mathbf{W}_s \in \mathbb{R}^{2H \times 1}$ is a learnable parameter.

The temporal attention weights are computed using softmax over the time dimension:

$$\boldsymbol{\alpha}^{(s)} = \text{softmax}(\mathbf{u}^{(s)}) = \frac{\exp(\mathbf{u}^{(s)})}{\sum_{t=1}^{T} \exp(\mathbf{u}_{:,t,:}^{(s)})}, \tag{8}$$

where $\boldsymbol{\alpha}^{(s)} \in \mathbb{R}^{N \times T \times 1}$. The context vector for symptom $s$ is computed as the weighted sum of hidden states:

$$\mathbf{c}^{(s)} = \boldsymbol{\alpha}^{(s)\top}\mathbf{H} \in \mathbb{R}^{N \times 2H}, \tag{9}$$

*4) Prediction Layer:* Each item or symptom-specific context vector $\mathbf{c}^{(s)}$ is passed through a shared fully connected network to produce predictions using nonlinear and linear transformations:

$$\hat{\mathbf{y}}_{\text{full}} = \text{ReLU}\big(\mathbf{c}^{(s)}\mathbf{W}_1 + \mathbf{b}_1\big)\mathbf{W}_2 + \mathbf{b}_2 \tag{10}$$

where $\mathbf{W}_1 \in \mathbb{R}^{2H \times H}$, $\mathbf{W}_2 \in \mathbb{R}^{H \times S}$, and $\mathbf{b}_1, \mathbf{b}_2$ are learnable parameters.

The final prediction for symptom $s$ is extracted as:

$$\hat{\mathbf{y}}^{(s)} = \hat{\mathbf{y}}_{\text{full}}[:, s] \in \mathbb{R}^{N}. \tag{11}$$

The complete prediction matrix is formed by stacking all symptom predictions:

$$\hat{\mathbf{Y}} = [\hat{\mathbf{y}}^{(1)}, \hat{\mathbf{y}}^{(2)}, \ldots, \hat{\mathbf{y}}^{(S)}] \in \mathbb{R}^{N \times S}. \tag{12}$$

For multi-step forecasting, predictions are generated recursively. Starting from baseline data $\mathbf{x}_{t_0}$, each predicted time step is appended to the input sequence:

$$\mathbf{X}_{t+1} = \text{concat}(\mathbf{X}_t, \hat{\mathbf{y}}_t), \tag{13}$$

where $\mathbf{X}_t$ is the input sequence up to step $t$.

*5) Model Interpretation:* The dual attention mechanism enhances interpretability via item-wise attention weights $\mathbf{A}$, highlighting key symptoms, and symptom-specific temporal attention weights $\boldsymbol{\alpha}^{(s)}$, identifying relevant historical time points per symptom.

Item attention weights represent each historical symptom's overall contribution to forecasts as percentages. Temporal attention weights indicate the contribution of each past time point to each predicted symptom, normalized to sum to 100%. All weights are aggregated across all $N$ patients.

### C. Training and Forecasting

We use five-fold cross-validation with an 80/20 train-test split, ensuring patients appear in only one test fold to avoid data leakage. Training folds include full longitudinal PRO sequences; test folds provide baseline data for forecasting future symptoms.

The model is trained end-to-end via backpropagation through time using the Adam optimizer and mean squared error loss with teacher forcing. Training stops after a set number of epochs or early stopping on a 20% validation split. To prevent overfitting, layer normalization, dropout, and L2 regularization are applied. Multi-step forecasts are generated recursively by feeding prior predictions as inputs, with outputs clamped to valid symptom ranges.

### D. Dataset Description

We evaluate our approach using a longitudinal PRO dataset from HNC patients treated at M.D. Anderson Cancer Center (MDACC), collected using the M.D. Anderson Symptom Inventory – Head and Neck module (MDASI-HN) [5]. This 28-item questionnaire assesses symptom severity on a 0–10 scale. Our analysis includes 12 general and 10 HNC-specific symptom items as illustrated in Figure 2 excluding the remaining 6 interference items to avoid multicollinearity [15].

PROs were recorded at 12 time points: baseline, start of radiotherapy (startRT), weekly during treatment (wk2–wk6), end of treatment (endRT), and post-treatment follow-ups at 6 weeks (wk6-post), 6 months (m6), and 12 months (m12). Forecast targets are set at 12 months (m12), corresponding to a clinically relevant endpoint for long-term outcome evaluation.

We included patients who rated each symptom at least once, and missing values in the dataset were imputed using symptom-based collaborative filtering, ensuring complete data for all subsequent analyses [10].

This study is covered under the MD Anderson Institutional Review Board (IRB) protocol RCR-003-0800. In compliance with the Health Insurance Portability and Accountability Act (HIPAA), informed consent was waived and approved by the IRB as all analyses were performed on retrospective anonymized data.

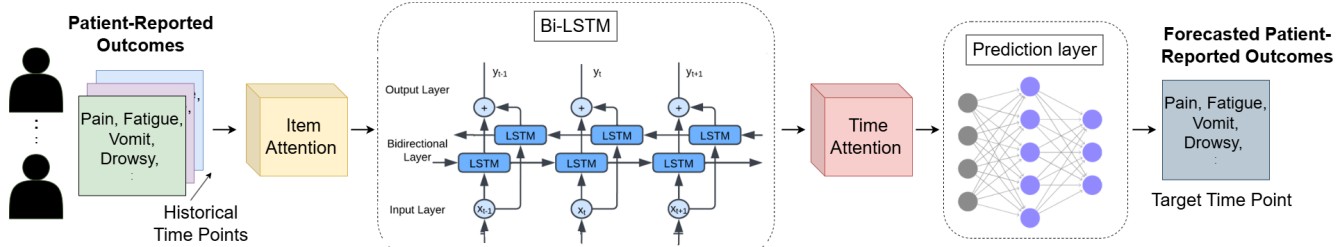

Fig. 1: Overview of the proposed attention-enhanced Bi-LSTM architecture for PRO forecasting with item and temporal attention mechanisms.

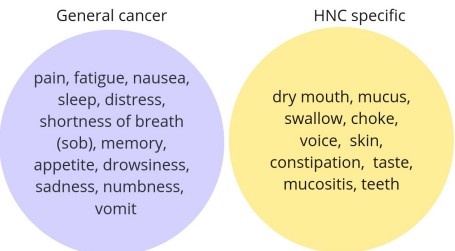

Fig. 2: Symptoms from the MDASI-HN questionnaire categorized into general cancer symptoms and head and neck cancer (HNC)-specific symptoms.

### E. Evaluation

Model performance at the target time point $t$ is evaluated using symptom-wise regression metrics: Root Mean Squared Error (RMSE) and Coefficient of Determination ($R^2$). RMSE captures prediction error magnitude, while $R^2$ assesses the model's ability to explain symptom variance, with values closer to 1 indicating better performance. Metrics are averaged across 22 symptoms and all five test folds, with 95% confidence intervals reported to assess stability.

$$\text{RMSE}_s = \sqrt{\frac{1}{N} \sum_{n=1}^{N} (\hat{y}_{n,s} - y_{n,s})^2}, \quad (14)$$

$$R_s^2 = 1 - \frac{\sum_{n=1}^{N} (y_{n,s} - \hat{y}_{n,s})^2}{\sum_{n=1}^{N} (y_{n,s} - \bar{y}_s)^2}, \quad (15)$$

where $N$ is the number of patients, $y_{n,s}$ and $\hat{y}_{n,s}$ are the true and predicted values for symptom $s$, and $\bar{y}_s$ is the mean ground truth value.

To assess the contribution of the dual-attention mechanism, we performed an ablation study comparing the proposed DAt-BiLSTM with a standard BiLSTM lacking item and temporal attention. Both models share identical architecture and training settings apart from the attention modules. Evaluation of both the BiLSTM and DAt-BiLSTM follows the same cross-validation protocol.

All models are implemented in PyTorch, with data processing and visualization handled using scikit-learn, NumPy, and Matplotlib.

## IV. RESULTS

### A. Data Summary

The MDASI-HN PRO cohort included a total of 821 HNC patients treated at MDACC, with a median age of 60 years. The majority were male (88.8%), and most tumors were located in the oropharyngeal region specifically, the base of the tongue or tonsils (86.4%). Based on the American Joint Committee on Cancer (AJCC) 8th edition staging criteria, 28.3% of patients presented with advanced primary tumors (T2–T4), while 50.2% exhibited advanced nodal disease (N2–N3). All patients underwent radiotherapy. In addition, 56.4% received concurrent chemotherapy, 19.4% received induction chemotherapy, and 19.4% underwent neck dissection surgery.

### B. Model Performance Comparison

Across all 22 symptoms and five test folds, the standard BiLSTM achieved an average RMSE of 1.69 [95% CI: 1.62–1.76], while the proposed DAt-BiLSTM yielded a lower RMSE of 1.62 [95% CI: 1.54–1.71]. This reflects improved accuracy by the DAt-BiLSTM in forecasting symptom values at the month 12 target time point.

Regarding explained variance, the DAt-BiLSTM achieves a mean $R^2$ score of 0.36 [95% CI: 0.19, 0.53], outperforming the BiLSTM's score of 0.26 [95% CI: 0.05, 0.46] across the five folds. This improvement suggests that DAt-BiLSTM is better able to account for variability in long-term symptom outcomes.

Focusing on RMSE for a detailed symptom-level comparison, Figure 3 shows the forecasting performance of the standard BiLSTM and the proposed DAt-BiLSTM models for month 12 PRO scores across all 22 symptoms. The results are reported on the combined test set formed by merging all five mutually exclusive folds.

The DAt-BiLSTM demonstrates better predictive performance, outperforming the standard BiLSTM on nearly all symptoms except for shortness of breath (sob) and choke. Overall, both models exhibit comparable performance with higher RMSEs for symptoms such as dry mouth and taste, while RMSEs for nausea, sob, vomiting, and skin problems are lower for both models.

Figure 4 shows boxplots comparing the predicted symptom ratings from both models against the original or ground truth

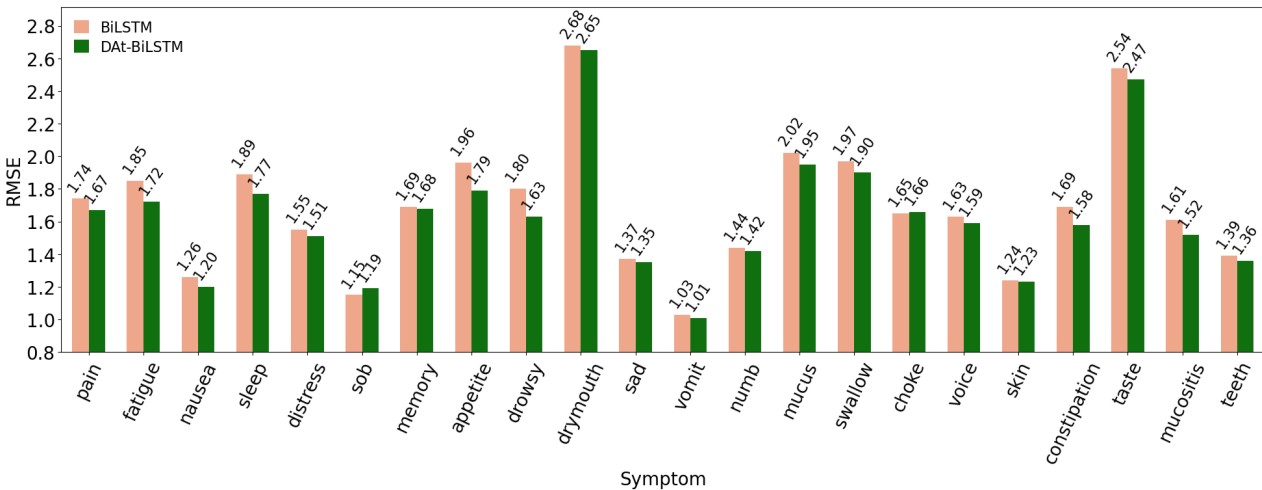

Fig. 3: Comparison of RMSE values for month 12 symptom predictions between the standard BiLSTM and the proposed DAt-BiLSTM models for all individual MDASI-HN symptoms.

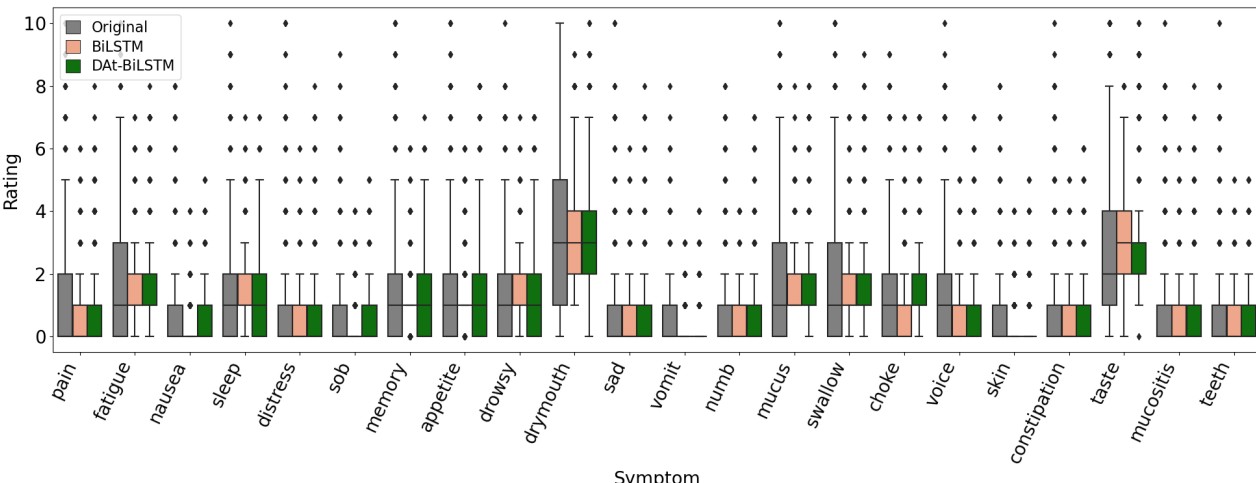

Fig. 4: Boxplots of original and predicted PRO scores at month 12 from the BiLSTM and DAt-BiLSTM models for all individual MDASI-HN symptoms.

ratings at month 12. The DAt-BiLSTM consistently produces distributions that more closely align with the ground truth, particularly for symptoms such as nausea, distress, memory, and constipation. Notably, even for sob and choke where the BiLSTM achieved marginally better RMSE, DAt-BiLSTM better approximates the underlying rating distribution, suggesting improved fidelity to symptom severity trends.

Furthermore, as illustrated in Figures 3 and 4, symptoms with higher prediction errors consistently correspond to those with higher severity levels, a relationship most clearly observed in dry mouth and taste.

Given that dry mouth and taste are among the most severe and challenging symptoms to forecast, we further assess each model's ability to approximate symptom severity thresholds rather than exact values, providing a further clinically meaningful evaluation of performance. Specifically, we binarize ratings using a threshold of 5, a commonly accepted cutoff

for identifying moderate to severe symptoms in the MDASI. Scores below 5 are labeled as non-severe (class 0), while scores of 5 or higher indicate moderate to severe symptoms (class 1). We then compute area under the receiver operating characteristic curve (AUC) and accuracy and compare confusion matrices between predicted and true ratings.

Figures 5a and 5b show the confusion matrices for dry mouth and taste, respectively, with BiLSTM results on the left and DAt-BiLSTM on the right in each figure. In both cases, the DAt-BiLSTM achieves slightly better classification performance. However, both models tend to underestimate the most severe symptom category, highlighting an important area for future improvement.

### C. DAt-BiLSTM Model Interpretation

To interpret the model's predictions, we examine the item and temporal attention weights learned by the DAt-BiLSTM.

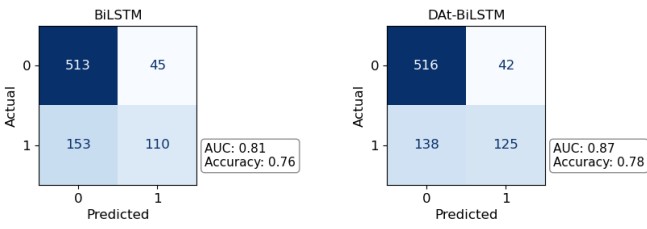

(a) Dry Mouth (Threshold $\geq$ 5)

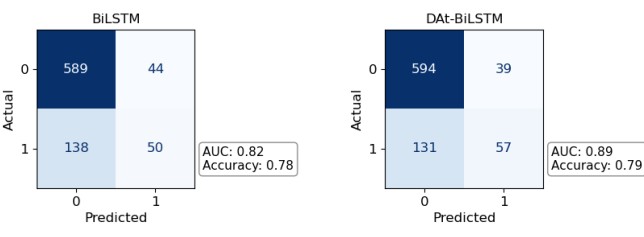

(b) Taste (Threshold $\geq$ 5)

Fig. 5: Confusion Matrices, AUC, and Accuracy for BiLSTM and DAt-BiLSTM Predictions of Threshold-5 Severity Levels for (a) Dry Mouth and (b) Taste Symptoms at Month 12.

Since the temporal attention operates on representations already refined by item-level attention, we present the temporal attention results first, followed by the item-level contributions.

*1) Temporal Attention Weights:* Figure 6 presents a heatmap illustrating the contribution of each historical time point to the prediction of symptoms at month 12. The results reveal that most predictions are strongly influenced by earlier time points particularly the baseline and startRT as well as later or post-treatment time points such as wk6_post and month 6 (m6).

Symptoms typically linked to HNC tumor characteristics, such as nausea, shortness of breath (sob), memory issues, vomiting, numbness, and skin issues, are primarily influenced by pre-treatment or early treatment time points.

Notably, more generalized symptoms such as pain, distress, sadness, and constipation show contributions from both pre- and post-treatment time points.

Symptoms often exacerbated by radiotherapy, including sleep disturbances, appetite loss, drowsiness, dry mouth, mucus, swallowing difficulties, and taste changes, are more heavily influenced by post-treatment time points, as expected.

*2) Item Attention Weights:* Figure 7 summarizes the contribution of individual symptoms across all historical time points in forecasting month-12 outcomes. The results indicate that symptoms such as dry mouth, appetite, sleep, and taste carry the highest predictive influence.

## V. DISCUSSION

This study shows that combining item and temporal attention in a Bi-LSTM framework improves both prediction and interpretability for long-term patient-reported outcomes (PROs) in head and neck cancer (HNC) patients. The DAt-BiLSTM model selectively weighs symptom-time point interactions, using the full temporal scope of PRO data to better forecast symptom severity at 12 months post-treatment.

Accurate symptom prediction is important for managing long-term toxicities. Most PRO analyses are retrospective and support reactive care. In contrast, our model uses early PRO data, especially at baseline, to anticipate future symptom trends. This enables more proactive treatment planning. Improvements in RMSE, $R^2$, and classification accuracy across multiple symptoms support its potential use in clinical monitoring, especially during survivorship when delayed toxicities affect quality of life.

The dual-attention mechanism helps the model focus on the most informative symptom-time combinations. Unlike standard Bi-LSTMs, which treat all inputs uniformly, our model gives more weight to inputs that enhance prediction. This improves tracking of symptoms such as nausea, shortness of breath, sadness, vomiting, and skin issues many of which affect treatment tolerance and long-term well-being.

Attention weights also reveal meaningful clinical patterns. Temporal attention shows that baseline and start-of-treatment time points are most predictive for tumor-related symptoms (e.g., nausea, shortness of breath, vomiting, skin), which often improve as treatment begins. In contrast, later time points (week 6 post-treatment and month 6) are more predictive for symptoms linked to treatment toxicity (e.g., sleep issues, appetite loss, dry mouth, drowsiness, mucus, swallowing, and taste), which tend to worsen after therapy ends. These trends align with known HNC symptom trajectories and highlight the value of the temporal interpretation provided by the model.

Item-level attention points to dry mouth, appetite, sleep, and taste as key symptoms for forecasting. This is consistent with previous research showing these symptoms often signal long-term toxicity [15], [7]. Their importance in the model suggests they should be closely monitored and managed during and after treatment.

Although the model performs well overall without including certain clinical factors such as tumor location, comorbidities, or radiation dose, adding these could improve the prediction of severe symptoms such as dry mouth and taste changes. Class imbalance may also contribute to higher errors in severe cases; techniques such as focal loss or oversampling could help address this.

Additionally, while attention improves interpretability by highlighting relevant inputs, it reflects correlation rather than causation and should support, not direct, clinical decisions. Finally, although our dataset is from a well-established source, external validation is necessary, as differences in patient populations, treatment protocols, and clinical settings may impact generalizability.

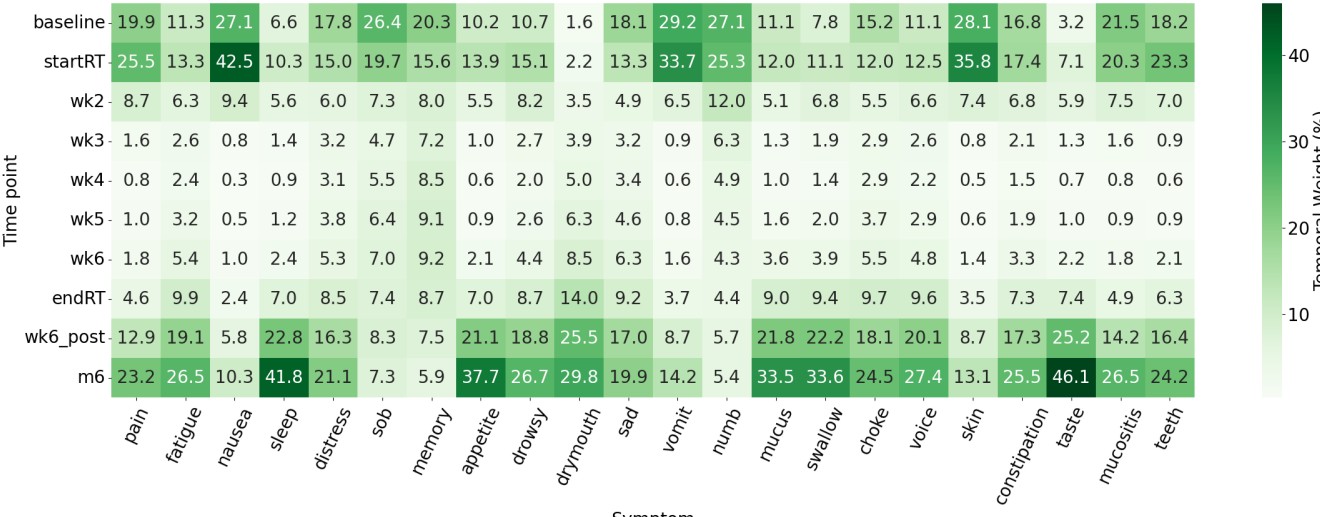

Fig. 6: Heatmap showing normalized attention weights assigned to historical time points for each predicted target symptom, where higher values indicate greater influence of the earlier time point on the month 12 forecast.

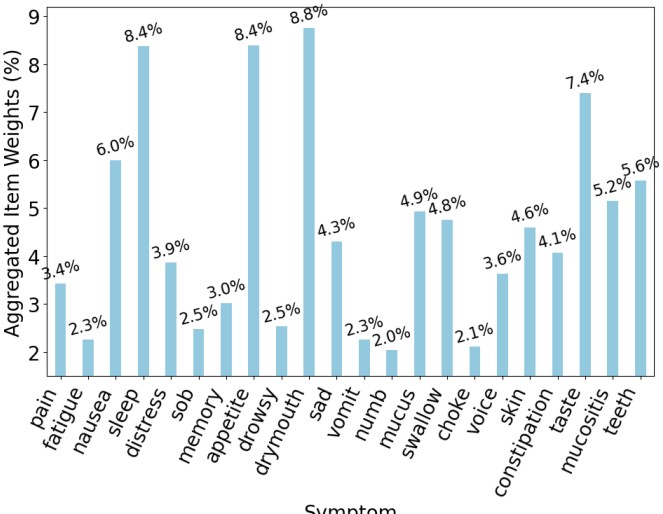

Fig. 7: Item attention weights across the 22 PRO symptoms, highlighting the relative importance of each symptom in the model's predictions.

In summary, DAt-BiLSTM improves long-term symptom forecasting from early PROs, supporting proactive oncology care for HNC patients.

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
