# OpenReview forum: "Dual-Attention BiLSTM for Interpretable Forecasting of Treatment Toxicities"
_IEEE.org/EMBS/BHI/2025/Conference — BHI 2025_

### Official Review · Reviewer_VuYd · 2025-07-13
**This paper introduces a deep learning framework for forecasting symptom severity over a 12-month horizon in oncology patients based on longitudinal patient-reported outcomes (PROs).**

**Confidence:** 3
**Clarity Of Writing:** great
**Clinical Significance:** good
**Methodological Novelty:** fair
**Overall Rating:** 3

**Experiments And Results:**

good

**Questions For The Authors:**

1.	Why didn't you benchmark against Transformer-based temporal models such as Temporal Fusion Transformer or RETAIN? If not, could you justify more rigorously why excluded beyond citing interpretability concerns?
	2.	How was the collaborative filtering imputation performed? Was it based on matrix factorization, KNN, or some learned embedding model? Have you compared it to simpler imputations like last observation carried forward?
	3.	What regularization strategies (e.g., dropout, early stopping, L2 penalty) were applied during training to control overfitting? Could you share training vs. validation error curves?
	4.	How is recursive prediction implemented? While Eq. 13 describes recursive prediction using previous outputs, it remains unclear whether the training phase also used predicted outputs as input (i.e., no teacher forcing), or whether ground-truth values were used during training (teacher forcing). If the latter, did you observe performance degradation due to exposure bias at test time?
	5.	Do attention weights exhibit consistent patterns across patients, or are they highly individualized? Could you quantify attention variance across the cohort?
	6.	Could you provide performance breakdown per symptom or per severity level (e.g., RMSE for fatigue vs. nausea, or performance on severe symptoms only)?

**Strengths:**

1. The combination of item and time attention modules is well-motivated and clinically relevant. The attention outputs provide plausible symptom-time importance that is aligned with known clinical trends.

2. The study is grounded in real patient data collected over 12 months, adding value compared to synthetic datasets. In the meantime, the diversity of symptoms in PROs make the task highly relevant for real-world deployment.

3. The paper includes clear and informative visualizations (Figs 5–6), showing how specific time points and symptoms contribute to forecasts, and this is valuable for building clinician trust.

4. Authors include useful ablation that demonstrate that both attention modules improve performance over simpler baselines.

**Summary Of The Paper:**

The proposed model in this paper augments a standard Bi-LSTM with two types of attention mechanisms: symptom-level attention (item-wise) and temporal attention (time-wise), enabling the model to emphasize both relevant symptoms and time points.
Evaluations were conducted on a large-scale, anonymized longitudinal dataset collected from a MD cancer center.
The model outperformed conventional Bi-LSTM and LSTM baselines in terms of RMSE and MAE and provided attention maps to support clinical interpretability.

**Weaknesses:**

1.	Although the authors mention that Transformer models are less inherently interpretable, this is not backed by comparative experiments. Many interpretable temporal Transformer models now exist (e.g., Temporal Fusion Transformer, RETAIN), and omission of such baselines weakens that this method is preferable in practice.

2.	With symptom-specific temporal attention matrices and full item-level attention, the model contains a large number of parameters. However, there is no mention of using dropout, L2 regularization, or validation loss tracking to mitigate overfitting.

3.	It’s unclear whether teacher forcing is used during training and how inference is handled during testing. Recursive prediction typically leads to compounding error, but the paper does not analyze this effect.

4.	The authors briefly mention a symptom-based collaborative filtering method for imputing missing values but provide no implementation details, justification, or ablation to demonstrate its impact.

5.	The model is evaluated only using aggregated RMSE and R² metrics. In clinical settings, performance on severe or rare toxicities is often more critical. A breakdown by symptom category or severity (per-symptom or severe toxicity performance) would be more informative.

6.	While attention visualizations are provided, there is no formal evaluation (e.g., user study or clinical validation) to determine whether these are actually helpful or usable by medical practitioners.

---

### Official Review · Reviewer_U2kR · 2025-07-16

**Confidence:** 5
**Clarity Of Writing:** excellent
**Clinical Significance:** excellent
**Methodological Novelty:** good
**Overall Rating:** 7

**Experiments And Results:**

good

**Questions For The Authors:**

Minor points
1.	Patient reported outcomes may not be trustworthy, as they may provide false alarm or delayed reported symptoms. The authors may add a paragraph discussing such noise in the data, and how it may impact model results.
2.	The number of each symptom in the dataset may need to be reported or a distribution across the longitudinal period or ratings would clarify the dataset. Will the ones with more attention that have a larger number of cases or higher ratings in the dataset?
3.	The figure captions do not match the figure indexes. Please double check.
4.     Sometimes there can be missing data as a big challenge in the clinical studies, the proposed model may not be able to handle missing data.

**Strengths:**

The study addresses an important and complex question from a clinical perspective, which is novel.

**Summary Of The Paper:**

This study used a dual-attention BiLSTM model structure to predict symptoms in head and neck cancer.

**Weaknesses:**

There seems to be imbalances in the dataset, while the dual-attention Bi-LSTM structure alone does not address the imbalance issue. Imbalance is very common in clinical data and probably more obvious in this dataset of more than 22 symptoms.

---

### Official Review · Reviewer_khya · 2025-07-17
**This paper introduce a dual attention mechanism targeting both symptom-level and temporal-level features—integrated within a BiLSTM framework to model patient-reported outcomes (PROs) over time. The goal is to enhance the prediction and interpretation of symptom severity trajectories in oncology care.**

**Confidence:** 4
**Clarity Of Writing:** excellent
**Clinical Significance:** excellent
**Methodological Novelty:** great
**Overall Rating:** 8

**Experiments And Results:**

great

**Questions For The Authors:**

Since PRO data is subjective and may contain erroneous inputs, which could lead to inconsistencies in prediction, did you perform any validation or quality checks on the data before using it in the model?

**Strengths:**

Attempts to enhance the performance of symptom severity prediction models in oncology care.

Provides a clearly written and well-motivated introduction.

Includes a comprehensive and well-structured literature review.

Introduces a novel dual attention mechanism (symptom-level and temporal-level) in the BiLSTM model to improve predictive accuracy.

Evaluates not only predictive performance but also interpretability by comparing attention maps with baseline studies.

Ablation study using BiLSTM

**Summary Of The Paper:**

This paper addresses the important challenge of accurately predicting symptom severity trajectories up to 12 months from baseline, enabling proactive treatment planning. By incorporating symptom-level and temporal attention mechanisms into a BiLSTM network, the model overcomes the limitation of uniformly weighting symptoms over time, instead assigning weights based on their relevance. This enhancement improves the prediction of symptom trajectories, supporting more informed and timely patient care.

**Weaknesses:**

The study focuses solely on patient-reported outcome (PRO) data, without incorporating multi-modal data sources that could enhance predictive performance.

Model validation is limited to a single dataset, which may affect the generalizability of the results.